# Models of Follow-Up Care and Secondary Prevention Measures for Survivors of Colorectal Cancer: Evidence-Based Guidelines and Systematic Review

Jacqueline Galica [1,*], Caroline Zwaal [2,*], Erin Kennedy [3], Tim Asmis [4], Charles Cho [5], Alexandra Ginty [6] and Anand Govindarajan [3]

1 Queen's University Cancer Research Institute, Division of Cancer Care and Epidemiology, Queen's University, Kingston, ON K7L 3N6, Canada
2 Program in Evidence-Based Care, McMaster University, Hamilton, ON L8S 4L8, Canada
3 Mount Sinai Hospital, Toronto, ON M5G 1X5, Canada; erin.kennedy@sinaihealth.ca (E.K.); anand.govindarajan@sinaihealth.ca (A.G.)
4 Ottawa Hospital Cancer Centre, The Hospital General Campus, Ottawa, ON K1H 8L6, Canada; tasmis@toh.ca
5 Southlake Regional Cancer Centre, Newmarket, ON L3Y 2P9, Canada; ccho@southlakeregional.org
6 Dorval Medical FHT, Oakville, ON L6K 3W6, Canada; AGINTY@dorvalmedical.ca
* Correspondence: jacqueline.galica@queensu.ca (J.G.); zwaalc@mcmaster.ca (C.Z.)

**Abstract:** Objective: To provide recommendations for preferred models of follow-up care for stage I–IV colorectal (CRC) cancer survivors in Ontario; to identify signs and symptoms of potential recurrence and when to investigate; and to evaluate patient information and support needs during the post-treatment survivorship period. Methods: Consistent with the Program in Evidence-Based Medicine's standardized approach, MEDLINE, EMBASE, PubMed, Cochrane Library, and PROSPERO databases were systematically searched. The authors drafted recommendations and revised them based on the comments from internal and external reviewers. Results: Four guidelines, three systematic reviews, three randomized controlled trials, and three cohort studies provided evidence to develop recommendations. Conclusions: Colorectal cancer follow-up care is complex and requires multidisciplinary, coordinated care delivered by the cancer specialist, primary care provider, and allied health professionals. While there is limited evidence to support a shared care model for follow-up, this approach is deemed to be best suited to meet patient needs; however, the roles and responsibilities of care providers need to be clearly defined, and patients need to know when and how to contact them. Although there is insufficient evidence to recommend any individual or combination of signs or symptoms as strong predictor(s) of recurrence, patients should be educated about these and know which care provider to contact if they develop any new or concerning symptoms. Psychosocial support and empathetic, effective, and coordinated communication are most valued by patients for their post-treatment follow-up care. Continuing professional education should emphasize the importance of communication skills and coordination of communication between the patient, family, and healthcare providers.

**Keywords:** colorectal cancer; surveillance; follow-up; survivorship; support needs

## 1. Introduction

Approximately 26,900 Canadians are diagnosed with colorectal cancer in a single year (Canadian Cancer Society) [1]. With advancements in screening, diagnosis, and treatment, there has been a steady increase in the number of long-term (≥5 years following diagnosis) colorectal cancer (CRC) survivors [2]. With these increasing numbers, a greater focus should be directed towards ascertaining the best model of follow-up care for CRC survivors. An evidence-based follow-up care model reflecting current best practices may help healthcare providers make important care decisions and offer guidance on various aspects of clinical

management, such as who should perform patient follow-up (i.e., medical oncologist, radiation oncologist, surgeon, nurse practitioner, physician assistant, or family physician). However, there have been limited recommendations for a preferred model of follow-up care for patients with CRC to date; therefore, it is crucial that the best available evidence is determined.

The five-year recurrence rate for patients having had curative surgery for CRC is between 20–30% [2] and therefore monitoring for recurrence is an important aspect of their follow-up care. Recurrence may occur either locally or metastasize to other organs, most commonly the liver and/or lungs. However, the signs and symptoms of CRC recurrence may be subtle and difficult to determine, as they depend on the site of recurrence and may largely vary between patients. For this reason, both clinicians and survivors should be aware of signs and symptoms associated with CRC recurrence as an important aspect of follow-up care. As such, synthesizing this vast literature base would be useful in this regard.

In addition to the formerly identified important features of follow-up care, it is recommended that CRC survivors receive greater psychosocial support and communication with their healthcare providers [3]. As such, it is imperative that their follow-up care is based upon the individual needs of survivors, including their functional, physical, and psychosocial concerns, which may last for months or years after treatment [4]. Clinician and patient awareness of these long-term and late effects may help mitigate discomfort, effectively manage symptoms, and improve the overall quality of life. A summary of the patients' informational and support needs of patients would be useful to inform post-treatment clinical discussions.

Given the increasing number of long-term colorectal cancer survivors and the features essential to include in their follow-up care, the Program in Evidence-Based Care (PEBC) of Ontario Health (Cancer Care Ontario) (OH [CCO]) worked with Ontario stakeholders to develop an evidence-based guideline using the methodologies of the Practice Guidelines Development Cycle [5,6] and the AGREE II framework [7]. The purpose was to update the previous Ontario Health guideline [8]. In alignment with this process, the systematic review reported herein was conducted to determine: (i) the optimal model of care for follow-up and surveillance for those who have completed treatment for CRC; (i.e., should patient follow-up be done by a medical oncologist, radiation oncologist, surgeon, nurse practitioner, physician assistant, or family physician); (ii) signs and symptoms that may be predictive of a CRC recurrence; and (iii) post-treatment information and support needs of CRC survivors. The full guideline can be found at: https://www.cancercareontario.ca/en (accessed on 5 September 2020).

*Research Questions*

1.  Are there optimal models of follow-up care for persons who have completed treatment for CRC (i.e., which healthcare professionals should conduct patient follow-up?)
2.  What are the signs and/or symptoms that may signify a potential recurrence of CRC and therefore warrant more investigation?
3.  What are patients' post-treatment informational and support needs regarding their risk of recurrence and common long-term and late effects of CRC?

Intended users of this guideline include clinicians (e.g., medical oncologists, radiation oncologists, surgeons, advanced practice nurses, physician assistants, primary care providers (family physicians, nurse practitioners, family practice nurses)) involved in the delivery of care for colorectal cancer survivors. As well, this guideline could be utilized by healthcare organizations and system leaders responsible for offering, monitoring, or providing resources for colorectal cancer survivorship protocols.

## 2. Materials and Methods

### 2.1. Literature Search

First, a search for evidence guidelines and then a search for systematic reviews and primary literature was conducted. Being that this guideline is an update, the search date

was based on the previous guideline's dates, and the search terms were similar to the original guideline. On 8 March 2019, the search terms 'colorectal cancer', 'follow-up', 'surveillance', and 'survivors' were used to search for guidelines in the following sources: American Society of Clinical Oncology, Canadian Medical Association Journal Infobase, and National Institute for Health and Care Excellence Evidence Search, National Health and Medical Research Council–Australia Clinical Practice Guidelines Portal, and Cancer Council Australia–Cancer Guidelines Wiki. Evidence-based guidelines with systematic reviews that addressed at least one research question were included and if the guideline had a score of 5/7 or above on the rigor of development section of the AGREE II [7] and were published after 2016.

Since no guidelines were deemed fully endorsable, a search was conducted for existing systematic reviews on 1 May 2019, and for primary literature on 5 June 2019. The databases searched were OVID MEDLINE, EMBASE, and the Cochrane Database of Systematic Reviews for the years 2011 to 2019 (See Appendix A for search terms). Systematic reviews were included if they were in English and were relevant to the research questions. An update for the literature search was completed September 2020. Primary articles were included if they were randomized controlled trials, retrospective and prospective cohort studies with at least 30 participants, comparative cohort with at least 30 participants per group, with a minimum follow-up of two years and the population consisted of patients with CRC whose primary treatment was with curative intent and were without evidence of disease. Articles were excluded if they were letters, comments, editorials, non-English publications, abstracts or published before 2011. This systematic review has been registered on the PROSPERO (International prospective register of systematic reviews) website with the registration number CRD42020132109.

A review of the titles and abstracts was conducted by CZ. For studies that warranted full-text review, CZ reviewed each study independently and verified with another reviewer (EK) if uncertainty existed. All reviews and primary studies that met the inclusion criteria underwent data extraction by CZ, with all extracted data and information audited subsequently by an independent auditor (FM).

Assessment of systematic reviews was completed using the Risk of Bias in Systematic Reviews (ROBIS) tool [9]; RCTs via the Cochrane Risk of Bias tool [10]; and all non-RCTs using the Cochrane Risk of Bias in Non-Randomized Studies of Interventions (ROBINS-I) tool [11]. The Grading of Recommendations, Assessment, Development, and Evaluations (GRADE) framework [12] was used to evaluate the certainty of the evidence, taking into account the risk of bias, inconsistency, indirectness, imprecision, and publication bias.

### 2.2. Internal and External Review

The internal review included an evaluation of the guideline by the Guideline Development Group Expert Panel and the PEBC Report Approval Panel. A Patient Consultation Group, consisting of patients, caregivers, or family members reviewed the recommendations and provided feedback on their comprehensibility, appropriateness, and feasibility. Then, external feedback was obtained from content experts and target users. First, a targeted peer review where individuals with content expertise identified by the Guideline Development Group were asked to review and provide feedback to the guideline document. Second, relevant care providers and other potential users of the guideline provided feedback on the guideline recommendations via an online survey. The results of the incorporation of this process into the final guideline are described in Section 3.2.

### 3. Results

### 3.1. Literature Search Results

In total, 22 guidelines were found. Of those, 17 did not meet the inclusion criteria leaving four guidelines relevant to the research questions. In the searches for systematic reviews and primary studies, 3830 articles were retrieved, of which 388 were included in the full-text review. There were 25 systematic reviews considered for full-text review and

three met the inclusion criteria and were relevant to the scope of the guideline. There were 388 primary studies that underwent full-text review, seven of which were retained. See Figure 1 for the PRISMA diagram and Appendix B for study characteristics and summary of results.

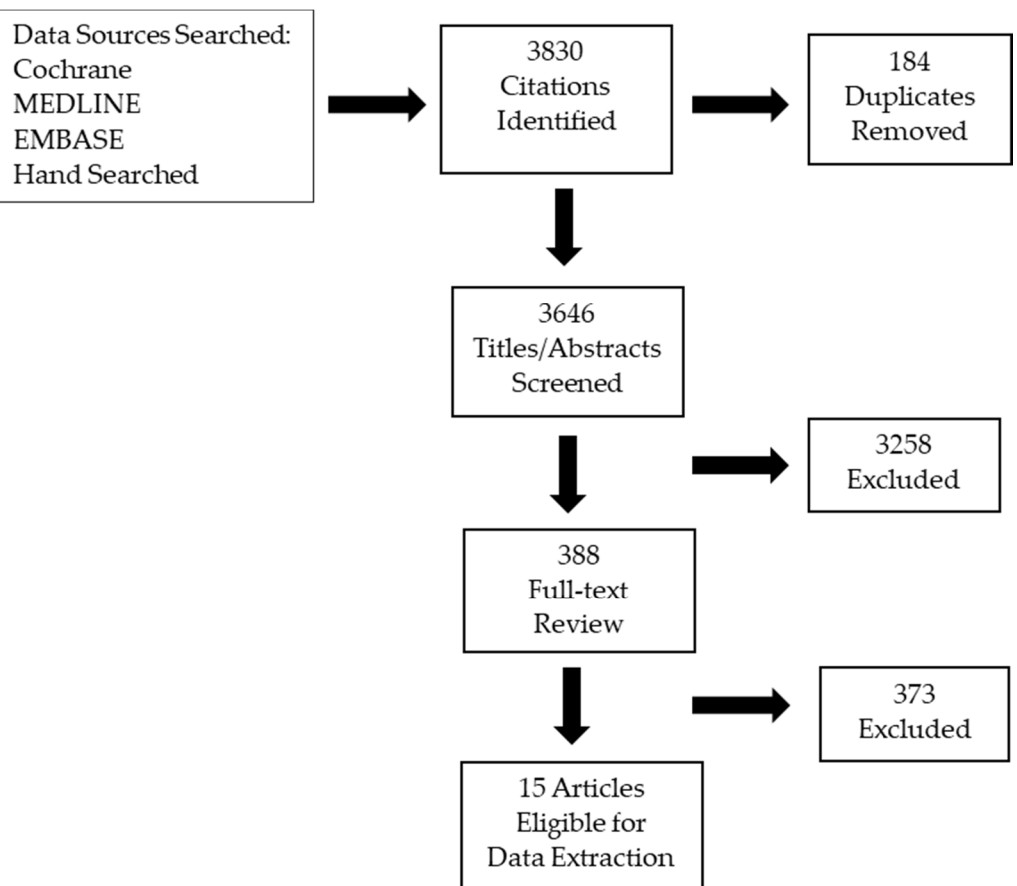

**Figure 1.** PRISMA Flow Diagram.

Appendix C Tables A5–A8 describe the results of the quality assessments. Overall, the risk of bias was considered to be low for each systematic review, low for the RCTs, and moderate for the cohort studies.

### 3.2. Internal and External Review

The Report Approval Panel members and Guideline Development Group Expert Panel approved the document outlining the results described from processes of Section 3.1. However, comments from the Expert Panel and Report Approval Panel reflected the need to clarify recommendations and to the list of long-term and late treatment efforts. The Patient Consultation Group supported the patient-focused recommendations and suggested that patients' families be included in communication recommendations.

Five targeted peer reviewers from Ontario, British Columbia and New York state reviewed the document. Comments included additional clarity regarding the surveillance of individual patients. Of the online survey sent to intended users of the guideline ($n = 182$), thirteen responses (7.1%) were received. Comments included a need for additional clarification and concision regarding the recommendations. Final guidelines recommendations reflect the integration of feedback obtained through both internal and external review processes.

## 4. Recommendations, Key Evidence and Interpretation of the Evidence

**Recommendation 1.** *Models of Follow-up Care.*

- *Follow-up care is complex and requires multidisciplinary, coordinated care of the patient delivered by the cancer specialist, family physician or nurse practitioner, and allied health professionals.*
- *The roles and responsibilities of the multidisciplinary team members need to be clearly defined and the patient needs to know when and how to contact each member of the team.*

*Evidence: Preferred models of follow-up care*

Three of the four retrieved guidelines had recommendations regarding models of follow-up care that were based upon a combination of selected evidence and consensus [8,13,14]. All guidelines recommend a combination of follow-up from care providers. The original OH (CCO) guideline recommendation acknowledged that the specialist-coordinated care within an institution is the most common practice for follow-up care in Ontario. However, they state that leaving specialist-led care and moving to family physician-coordinated or registered nurse (RN)-coordinated care are reasonable options [8]. Similarly, the Cancer Council Australia (CCA) colorectal cancer guideline concludes that follow-up care can be delivered as a combination of visits to the surgeon or associated gastroenterologist, with ongoing care by the family physician and clinical RN consultant [13]. National Comprehensive Cancer Network (NCCN) guidelines promote the clear delineation of oncologist and the primary care provider roles during the surveillance period [14].

The evidence about the optimal model of follow-up care was found in two systematic reviews [2,15], two RCTs [16,17], and two cohort studies [18,19]. Study outcomes were compared by provider, which in these studies included general practitioner (GP), registered nurse (RN), specialist, and nurse specialist. However, in this document we use the terms family physician (FP) and nurse practitioner (NP) to reflect current practice in the province of Ontario. Systematic review evidence showed no difference in overall survival between FP- or NP-led follow-up in the community compared with follow-up conducted in hospitals. In the two RCTS that compared FP or NP-led care to a surgeon, there was no difference in the recurrence rates of CRC. However, the sample sizes of patients and clinicians in these studies were small [16,17]. Adherence to guidelines was higher among nonphysician clinicians and FPs than surgeons. See Appendix B Tables A1 and A2 for Study Characteristics.

In the studies that examined quality of life and CRC follow-up, patients indicated that follow-up was important to them. Although patient satisfaction was high for all providers, their provider preference was dependent on their symptoms and individual needs without a clear preference for the type of provider (see Appendix B Tables A3 and A4 for comparisons).

**Recommendation 2.** *Signs and symptoms of potential recurrence*

- *The signs and symptoms of recurrence may be subtle or asymptomatic and must be considered in the context of the patient's overall health and pre-existing conditions. There is insufficient evidence to recommend any individual sign or symptom or combination of signs and symptoms as a strong predictor of recurrence.*
- *Patients should be educated about the potential signs and symptoms of CRC recurrence (see Table 1) and know which member of the multidisciplinary care team they should contact if they develop any new or concerning signs or symptoms.*

**Table 1.** Signs and Symptoms of Potential Recurrence.

| Sign or Symptom [1] | Type of Recurrence [2] | |
| --- | --- | --- |
| | **Local** | **Distant** [3] |
| Abdominal pain | X | X |
| Dry cough | | X |
| Rectal bleeding | X | |
| Changes in bowel habit | X | |
| Fatigue | X | X |
| Nausea | X | X |
| Unexplained weight loss | X | X |
| Anemia | X | X |
| Pain | X | |
| Stoma bleeding | X | |
| Palpable mass | X | X |
| Abdominal pain from hepatomegaly | | X |
| Jaundice | | X |
| Pleuritic chest pain or shortness of breath | | X |
| Anorexia, cachexia, and weight loss | | X |
| Dyspnea | | X |
| Loss of appetite | | X |
| Signs and/or symptoms specific to rectal cancer | | |
| Pelvic pain | X | |
| Sciatica | X | |
| Difficulty with urination or defecation | X | |

[1] There are no signs or symptoms specific to colon cancer that would not also apply to rectal cancer. [2] Both local and distant recurrence are most likely to occur in the first two years following treatment [20]. [3] Signs and symptoms have been categorized into those signs most commonly associated with local recurrence or distant metastasis (i.e., liver and/or lung metastasis) based on best available evidence and expert opinion.

### *Evidence: Signs and symptoms of potential recurrence*

The evidence for this recommendation comes primarily from the former OH (CCO) follow-up guideline, the Cancer Care Australia guideline, one RCT, and one retrospective study [8,13,21,22]. In the previous OH (CCO) guideline, common signs and symptoms associated with CRC recurrence were based upon expert opinion and included: abdominal pain, particularly in the right upper quadrant or flank (liver area), dry cough, and vague constitutional symptoms (i.e., fatigue, nausea, and unexplained weight loss) [8]. Specific to rectal cancer, pelvic pain, sciatica, and difficulty with urination or defecation were identified in the previous guideline.

The Cancer Care Australia guideline reported that for symptomatic patients, the symptoms will depend on local versus distant recurrence [13]. Local recurrences may include both anastomotic or luminal recurrences and symptoms may include rectal bleeding, anemia, altered bowel habits, or varying degrees of bowel obstruction. Patients with nodal or surgical bed recurrences may have a palpable mass or pain from a mass affecting neighbouring structures. In patients with rectal cancer with pelvic recurrences, pain is a common symptom. In distant or systemic recurrence, the most common sites are hepatic followed by pulmonary metastases. Symptoms vary depending on the site of recurrence and may include symptoms such as abdominal pain from hepatomegaly, jaundice, pleuritic chest pain, and shortness of breath. Patients with extensive disease may also have anorexia, cachexia, and weight loss.

In an RCT comparing surgeon versus family practitioner follow-up (*n* = 110 patients), Augested et al. (2013) found that 14 patients had cancer recurrence, seven of whom had symptoms [21]. In the retrospective cohort study by Duinveld et al. (2016), 74 of 446 patients (16.6%) had a recurrence, which was detected among 31 patients during a non-scheduled visit among whom 26 (84%) were symptomatic [22]. There were 38 local recurrences, of which 14 (37%) were symptomatic and 24 (63%) were asymptomatic. Among the 82 distant recurrences, 36 (44%) were symptomatic and 46 (56%) were asymptomatic.

**Recommendation 3.** *Common and/or substantial long-term and late effects*

- *Psychosocial support about the risk of CRC recurrence and provision of empathetic, effective, and coordinated communication are most highly valued by patients for post-treatment physical effects and symptom control.*
- *Continuing professional education should emphasize the importance of communication skills and coordination of communication between the patient and family, and healthcare providers. A list of late and long-term physical and psychosocial effects of CRC is found in Table 2 below.*

**Table 2.** Long-term and late effects.

| **Physical Long-term and Late Effects** |
| --- |
| <ul><li>Issues with bowel function<ul><li>Frequent and/or urgent bowel movements</li><li>Loose bowels</li><li>Incontinence</li><li>Gas and/or bloating</li></ul></li><li>Postoperative issues<ul><li>Possible but low risk of incisional hernia</li><li>Possible but low risk of bowel obstruction</li></ul></li><li>Peripheral neuropathy (associated with treatment using oxaliplatin)</li><li>Chemotherapy-related cognitive side effects</li><li>Issues with fertility</li><li>Sexuality function (e.g., vaginal dryness and pain with intercourse, erectile dysfunction, retrograde ejaculation)</li><li>Stoma care and lifestyle adjustments for patient who have received ostomy</li><li>Possible changes in urinary function</li><li>Chronic pain</li><li>Fatigue</li><li>Nutritional and diet considerations</li></ul> |

| **Psychosocial Long-term and Late Effects** | |
| --- | --- |
| <ul><li>Psychological distress</li><li>Depression</li><li>Anxiety</li><li>Worry</li><li>Fear of recurrence</li><li>Changes in sexual function/fertility</li></ul> | <ul><li>Body and/or self-image</li><li>Relationships</li><li>Other social role difficulties</li><li>Return to work concerns</li><li>Financial challenges</li><li>Support for family</li></ul> |

### *Evidence: Post-treatment informational and support needs for CRC survivors*

The evidence for this recommendation comes from two guidelines and one systematic review [13,23,24]. Five guidelines in the European Society of Coloproctology summary stated that structured preventive care with health-promoting initiatives should be part of supportive care provided to colorectal cancer survivors [23]. The Cancer Care Australia CRC guideline indicated that the provision of adequate information to patients with CRC is related to increased psychological well-being and that good communication skills are vital [13]. The group identified six main points regarding information that should be provided to colorectal cancer patients:

1. Clear explanation of treatments options along with potential effectiveness and adverse effects.
2. The physician should ensure that patients provide the amount of detail they prefer to receive and to enable the patients' desired amount of involvement in decision making.
3. Clinicians need to ensure that the patient understands the information, and their reactions in order to provide emotional support.
4. Clinicians need to provide written materials and should consider offering audio recordings of key consultations. The use of a specialist nurse or counsellor, a follow-up letter, and/or educational programs may also assist in recall of information.
5. Information should be made available over time and longer appointments that review information that allows for further integration could be scheduled.
6. Families and caregivers of patients should be kept informed of discussions and information.

According to a systematic review about the supportive care needs of CRC survivors, the highest priority supportive care needs are for information and education and physician communication, particularly around the risk of recurrence [24]. While this information was identified as important by patients, so was the way in which this information was provided to them in a coordinated, honest, unhurried, and empathetic approach. Though physical symptoms were important to know, they were not rated as highly as information, education, and physician communication [24].

### Evidence: Long-term and late treatment effects

Four guidelines and one systematic review identified 39 physical and psychosocial long-term and late effects of CRC [8,13,14,23,24]. These are summarized in Table 2.

### 5. Discussion

This systematic review provided a comprehensive examination about the optimal model of follow-up care for CRC survivors. Based upon the evidence reviewed, it is critical that shared models of care integrate and coordinate care among patients, families, and healthcare providers. Of paramount importance is that patients know which provider to contact for specific issues and how to contact that provider. Enhanced communication and role clarity among clinicians is also needed. Innovative strategies, such as virtual care, may be useful to facilitate the integration and coordination of care. Indeed, remote follow-up led to enhanced involvement of CRC patients in their own care [25]. However, it is important to acknowledge that a "one size fits all" shared care model is unlikely to be used uniformly across the province. As such, shared care models will need to be tailored for specific organizations, regions, and Ontario Health Teams based upon the particular resources available to them.

A second goal of this review was to identify possible signs and symptoms of CRC recurrence that warrant investigation. The evidence for signs and symptoms was collected from consensus recommendations and from guidelines and small studies. Given that only 35% to 50% of patients with CRC recurrence will present with obvious symptoms means that both local and distant recurrence can be subtle and complex to identify. As such, it is important that patients are aware of which signs and symptoms may indicate a possible recurrence and that new signs and symptoms are investigated in a timely manner.

The third goal of this review identified the needs and long-term and late effects for CRC survivors. These results are important for clinicians and patients to be aware of so that patient discomfort can be mitigated, their symptoms effectively managed, and quality of life promoted. While physical symptoms were important to know, survivors did not rate these as highly as information, education, and physician communication. Indeed, one of the most interesting findings of this review was that CRC survivors prioritized their informational and supportive needs during follow-up, particularly about their fear of recurrence. As such, it is important for clinicians to specifically discuss the risk of recurrence with patients at follow-up visits. Even more importantly, is that patients value the manner in which

their healthcare provider presents this information. Patients consistently emphasize the importance of coordinated, honest, unhurried, and empathetic delivery of information by their healthcare providers. Based on these findings, providers should consider ongoing professional development opportunities to continue to grow their communication skills. This participation should also be encouraged at an institutional and organizational level.

*Limitations*

The evidence reviewed herein indicated no difference in overall survival nor CRC recurrence between varying models of follow-up care. However, these studies were small and had a small number of clinicians in each of the trials [16,17]. While a shared care model is preferred, there was little information on which shared care model is most beneficial or how this should be implemented. There were very few studies that incorporated virtual care or remote follow-up as part of this model.

While patient informational and supportive needs were highly consistent across studies, the quality of evidence came primarily from cross-sectional surveys and therefore is subject to recall and response rate bias. There was also limited information on racial disparities in the quality of follow care.

## 6. Conclusions

Colorectal cancer follow-up care is complex and requires multidisciplinary, coordinated care delivered by the cancer specialist, primary care provider, and allied health professionals. While there is limited evidence to support a shared care model for follow-up, this approach is deemed to be best suited to meet patient needs; however, the roles and responsibilities of care providers need to be clearly defined, and patients need to know when and how to contact them. Although there is insufficient evidence to recommend any individual or combination of signs or symptoms as strong predictor(s) of recurrence, patients should be educated about these and know which care provider to contact if they develop any new or concerning symptoms. Psychosocial support and empathetic, effective, and coordinated communication are most valued by patients for their post-treatment follow-up care. Continuing professional education should emphasize the importance of communication skills and coordination of communication between the patient, family, and healthcare providers.

**Author Contributions:** J.G., C.Z., E.K., T.A., C.C., A.G. (Alexandra Ginty) and A.G. (Anand Govindarajan) contributed to study design, data collection, data interpretation, manuscript revision. J.G. and C.Z. drafted the first report. C.Z. conducted the literature search, figures and tables. All authors have read and agreed to the published version of the manuscript.

**Funding:** The PEBC is a provincial initiative of OH (CCO), supported by the Ontario Ministry of Health (OMH). All work produced by the PEBC is editorially independent from the OMH.

**Data Availability Statement:** The data presented in this study are available in the appendices.

**Acknowledgments:** The Colorectal Cancer Survivorship Guideline Development Group (GDG) would like to thank the following individuals for their assistance in developing the guideline: Sheila McNair, Emily Vella, Donna Maziak, Jonathon Sussman Eric Winquist, Lisa Del Giudice, Stan Feinberg, Sharlene Gill, Mary Smith and Alice Wei for providing feedback on draft versions. Faith Maelzer and Megan Smyth for conducting data review and audit. Sara Miller for copy editing.

**Conflicts of Interest:** The authors declare no conflict of interest.

## Appendix A. Literature Search Strategy
MEDLINE
exp colorectal neoplasms/
colorectal cancer:.mp.
rectal cancer:.mp.
CRC:.mp.

or/1–4 6. surveillance:.mp.

follow-up:.mp.

survivor:.mp.

prevent:.mp.

(late adj2 effect:).mp.

or/6–10

5 and 11

recurrence/

neoplasm recurrence, local/

15. recurren:.mp.

or/13–15

12 and 16

limit 17 to (english language and humans) 19. limit 18 to yr = "2011–current"

meta-analysis.pt.

meta-analy$.tw.

metaanal$.tw.

(systematic adj (review$1 or overview$1)).tw.

meta-analysis as topic/

or/20–24

cochrane.ab.

(cinahl or cinhal).ab.

embase.ab.

scientific citation index.ab.

bids.ab.

cancerlit.ab.

or/26–31 33. reference list$.ab.

bibliograph$.ab.

hand-search$.ab.

relevant journals.ab.

manual search$.ab.

or/33–37 39. selection criteria.ab.

data extraction.ab.

39 or 40 42. review.pt.

review literature as topic/

42 or 43

41 and 44

comment.pt.

letter.pt.

editorial.pt.

or/46–48

25 or 32 or 38 or 45

50 not 49

practice guideline/53. practice guideline$.mp.

52 or 53

51 or 54

19 and 55

19 not 49

(comment or letter or editorial or note or erratum or short survey or news or newspaper article or patient education handout or case reports or historical article).pt.

19 not 58

59 and 55

59 not 55 62. case series.mp.

61 not 62

59 not 62

EMBASE
exp colorectal cancer/or exp colorectal carcinoma/or exp colorectal tumor/or exp colorectal tumour/
colorectal cancer:.mp.
rectal cancer:.mp.
CRC:.mp.
or/1–4
surveillance:.mp.
exp follow-up/
after care/
long term care/
follow-up:.mp.
survivor:.mp.
prevent:.mp.
(late adj2 effect:).mp.
or/6–13
5 and 14
exp recurrent cancer/or exp recurrent disease/
recurren:.mp.
16 or 17
15 and 18
limit 19 to (human and english language)
limit 20 to yr = "2011–current"
exp meta-analysis/
((meta adj analy$) or metaanaly$).tw.
(systematic adj (review$1 or overview$1)).tw.
or/22–24
cancerlit.ab.
cochrane.ab.
embase.ab.
(cinahl or cinhal).ab.
scientific citation index.ab.
bids.ab.
or/26–31 33. reference list$.ab.
bibliograph$.ab.
hand-search$.ab.
manual search$.ab.
relevant journals.ab.
or/33–37
data extraction.ab.
selection criteria.ab.
39 or 40
review.pt.
41 and 42
letter.pt.
editorial.pt.
44 or 45
25 or 32 or 38 or 43
47 not 46
exp practice guideline
practice guideline$.tw.
49 or 50
48 or 51
21 and 52

## Appendix B. Study Characteristics

**Table A1.** Study Characteristics of Systematic Reviews.

| Study | Number of Studies | Topic | Results |
|---|---|---|---|
| Jeffery, 2019 [2] | 19 | Overall survival | RCTs that compared different healthcare professionals and found no differences in a subgroup analysis ($X^2 = 0.40$; $p = 0.53$; $I^2 = 0\%$) between FP or NP-led follow-up (2 studies) and hospital follow-up (13 studies). The overall effect on overall survival was similar (HR, 0.91; 95% CI, 0.80 to 1.03, $p = 0.14$). |
| Berian [15] | 16 | Patients' perceptions and expectations of routine surveillance | 5 studies showed a preference for specialist-led care; 4 studies found equivalent preference for NP- and specialist-led follow-up; 4 studies showed equivalent preference for specialist- and FP-led care 1 study showed strong preference for NP-led follow-up over specialist-led follow-up patients reported high satisfaction with follow-up and believed that continued follow-up was important for the detection of recurrence. preferences varied for a given type of provider to conduct follow-up surveillance, satisfaction was generally high regardless of provider. |
| Kotronoulas [24] | 54 studies | Supportive care needs of people living with and beyond CRC | Identified 136 individual needs were identified and classified into 8 conceptual domains that included: (i) physical and cognitive, (ii) psychosocial and emotional, (iii) family related, (iv) social, (v) interpersonal and intimacy, (vi) daily living, (vii) Information/education, and (viii) patient-physician communication |

**Table A2.** Study Characteristics of Follow-up Providers.

| Study | Provider Used/ Surveillance Person/ Schedule | Number of Patients | Median Observation (Months) | Overall Recurrence Rate (%) | Timeliness/ Compliance | Rate of Late Effects/ Metastases | Time to Recurrence | Quality of Life/Patient Satisfaction | Unannounced Follow-Ups |
|---|---|---|---|---|---|---|---|---|---|
| Augestad, 2013 [16] RCT | FPs Surgeons | 55 55 | 75% for 12 mos, and 52% for 24 mos | 10.9 14.5 | Response rate of 96% for QoL questionnaire | NA | 35 days 45 days (Reported as serious clinical event) | No significant effect on QoL main outcome measures; EORTC QLQ C-30 subscales reported significant effects in favour of FP follow-up | 3 4 (Number of metastases surgeries) |
| Strand, 2011 [17] RCT Rectal cancer patients | Surgeon NP | 56 54 | 36 | 0 | All patients completed the questionnaire | 7 8 Distant metastases | NA | Overall high patient satisfaction; VAS 9.4 for surgeon and 9.5 for NP | 4 surgeries for distant metastases, 9 received palliative chemotherapy |
| Coeburgh van den Braak, 2018 [18] Prospective | NPC No NPC | 394 287 | 34.3 for DFS; 67.9 for OS | 12.5 | Involvement of an NPC resulted in a higher adherence to follow-up (84.3 vs. 73.9%, $p = 0.001$) | NA | NA | NA | NA |

Abbreviations: CEA = carcinoembryonic antigen; CRC = colorectal cancer; DFS = disease-free survival; EORTC QLQ = European Organisation for Research and Treatment of Cancer Quality of Life Questionnaire; FP = family practitioner; mo = month; NA = not applicable; NPC = nonphysician clinician; OS = overall survival; RCT = randomized controlled trial; NP = specialist nurse practitioner.

**Table A3.** Summary of Primary Literature Results Between Follow-up Providers.

| Study | Outcome | FP or NP vs. Hospital | FP vs. Surgeon | NP vs. Surgeon |
|---|---|---|---|---|
| | **Overall survival** | | | |
| Jeffrey [2] | | No difference | | |
| | **Recurrence** | | | |
| | Mean time until diagnosis | | No difference | |
| Augestad [16] | Cancer recurrence | | No difference | |
| | Died by metastatic | | No difference | |
| Strand [17] | Metastatic cancer | | | No difference |
| | **QoL** | | | |
| | Overall QoL | | No difference | |
| | Role functioning | | FP better $p = 0.02$ | |
| Augestad [16] | Emotional function | | FP better $p = 0.01$ | |
| | Pain | | FP better $p = 0.01$ | |
| | False positives | | No difference | |
| | Hospital travels (+cost) | | FP better $p < 0.001$ | |
| | **Patient satisfaction** | | | |
| | Pt satisfaction | | | No difference |
| Strand [17] | Anxiety | | | No difference |
| | Sufficient time spent | | | No difference |
| | **Unannounced follow-ups** | | | |
| | Longer consultation time | | | NP longer $p = 0.001$ |
| Strand [17] | Blood samples | | | NP more $p = 0.003$ |
| | Radiological tests | | | No difference |
| | **Adherence** | | | |
| Augestad [16] | Healthcare contacts | | FP had more | |
| | Diagnostic tests | | FP had more | |
| Coeburgh vander Braak [18] | Scheduled surveillance | Hospital with dedicated NPC better $p = 0.001$ | | |
| | **Patient preference** | | | |
| Weildraaijer [19] | Pt preference | | No difference | |
| Berian SR, *n* = number of articles [15] | Pt preference | Preference for specialist led: $n = 5$  Preference for NP led over specialist: $n = 1$  Equivalent NP vs. specialist led: $n = 4$ | Equivalent specialist vs. FP led: $n = 4$ | |

Abbreviations: RP = family practitioner; NPC = nonphysician clinician; QoL = quality of life; NP = nurse practitioner; SR = systematic review.

**Table A4.** Study Characteristics for Signs and Symptoms.

| Study | Follow-Up Program Intensity | Number of Patients and Disease Type | Median Observation (Months) | Overall Recurrence and Time to Recurrence | Rate of Late Effects/ Metastases | Signs and Symptoms Associated with Risk of Recurrence (Number and %) |
|---|---|---|---|---|---|---|
| Duineveld, 2016 [22] Retrospective cohort | CEA testing every 3 to 6 months during the first 3 years and 6 months during the following 2 years; abdominal imaging every 6 months for first 2 years and annually for following 3 years | 446 93 (21%) stage I carcinoma, 176 (39%) stage II, 176 (39%) stage III; majority carcinoma of left colon (55%) | 34 | 74 pts (16.6%) 43 (58%), detected during a scheduled follow-up visit; 41 (95%) asymptomatic 31 (42%), found during non-scheduled interval visits; 26 (84%) of these patients were symptomatic Time to recurrence: 13.7 months | 9 lung metastases | Symptoms reported during interval visits leading to detection of recurrent disease Abdominal pain: 15 (57.7) Altered defecation: 11 (42.3) Weight loss: 6 (23.1) Pain in back of pelvis: 4 (15.4) Fatigue: 2 (7.7) Dyspnea: 2 (7.7) Loss of appetite: 2 (7.7) Other (including urine retention, hematuria or cough): 3 (11.6) >1 symptom: 14 (53.8) |
| Augested, 2014 [21] RCT | CEA testing and clinical exam every 3 months during the first 2 years and 6 months during the following 3 years; chest x-ray and liver ultrasound every 6 months for first 2 years and annually for following 3 years; colonoscopy at 1 and 4 years | 110 Dukes' stage A, B or C colon cancer | 24 | 14 pts (12.7%) 7 had symptoms 7 found during visit Time to Recurrence: 45 days in surgeon group and 35 days in the GP group ($p$ = 0.46) | 48 serious clinical events (SCE; episode leading to suspicion of cancer recurrence) | Of 48 SCEs; 31 (65%) were initiated by emerging symptoms 17 (35%) were initiated by test findings. 14 pts had true colon cancer recurrence. 7 pts had symptoms: Abdominal pain-4 Blood in stool-1 Weight loss-1 Stoma bleeding-1 7 pts had radiologically detected lesions ($n$ = 4) and elevated CEA levels ($n$ = 3) |

## Appendix C. Quality Assessment Scores

**Table A5.** AGREE II—Guidelines.

| Guideline | Domain 1: Scope and Purpose | Domain 2: Stakeholder Involvement | Domain 3: Rigor of Development | Domain 4: Clarity of Presentation | Domain 5: Applicability | Domain 6: Editorial Independence |
|---|---|---|---|---|---|---|
| OH (CCO) [8] | 100% | 58.3% | 75% | 83.3% | 18.7% | 83.3% |
| ESC [19] | 95.2% | 42.8% | 78.5% | 85.7% | 28.5% | 78.5% |
| NCCN-colon [14] | 75% | 61.1% | 67.7% | 69.4% | 66.7% | 83.3% |
| CCA [13] | 95.2% | 90.4% | 85.7% | 71.4% | 60.7% | 85.7% |

Abbreviations: CCA = Cancer Council Australia; ESC = European Society of Coloproctology; NCCN = National Comprehensive Cancer Network; OH (CCO) = Ontario Health (Cancer Care Ontario).

**Table A6.** ROBIS—Systematic Review/Meta-analysis.

| Study | Domain 1: Study Eligibility Criteria | Domain 2: Identification and Selection of Studies | Domain 3: Data Collection and Study Appraisal | Domain 4: Synthesis and Findings | Overall Risk of Bias |
|---|---|---|---|---|---|
| Jeffery, 2019 [2] | Low | Low | Low | Low | Low |
| Berian, 2017 [15] | Low | Low | Low | Low/unclear | Low |
| Kotronoulas, 2017 [24] | Low | Low | Low | High | Low |

**Table A7.** Risk of Bias—RCTs.

| Study | Domain 1: Randomization Process | Domain 2: Deviation from Intervention | Domain 3: Missing Outcome Data | Domain 4: Measurement of Outcome | Domain 5: Reported Result | Overall Risk of Bias |
|---|---|---|---|---|---|---|
| Augestad, 2013 [16] | Low | Low | Low | Low | Low | Low |
| Strand, 2011 [17] | Low | Low | Low | Low | Low | Low |
| Augestad, 2014 [21] | Low | Low | Low | Low | Low | Low |

Abbreviations: RCTs = randomized controlled trials.

**Table A8.** Risk of Bias—Cohort Studies.

| Study | Domain 1: Bias Due to Confounding | Domain 2: Bias Due to Selection of Participants | Domain 3: Bias in Measurement of Interventions | Domain 4: Bias Due to Departure of Interventions | Domain 5: Bias Due to Missing Data | Domain 6: Bias in Measurement of Outcomes | Domain 7: Bias in Selection of the Reported Results | Overall Risk of Bias |
|---|---|---|---|---|---|---|---|---|
| Coebergh van den Braak, 2018 [18] Prospective | Moderate | Moderate | Moderate | Low | Moderate | Moderate | Moderate | Moderate |
| Wieldraaijer, 2018 [19] Retrospective | Moderate | Low | Low | Moderate | Low | Low | Low | Moderate |
| Duinveld, 2016 [22] Retrospective | Low | Low | Low | Low | Low | Low | Low | Low |

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
