# Peer review of "Models of Follow-Up Care and Secondary Prevention Measures for Survivors of Colorectal Cancer: Evidence-Based Guidelines and Systematic Review"

_curroncol, doi:10.3390/curroncol29020040_

Round 1

Reviewer 1 Report

Manuscript entitled "Models of Follow-up Care and Secondary Prevention Measures for Survivors of Colorectal Cancer: Recommendations and Systematic Review."

This work is of interest and of significance. Some issues should be improved:

  1. The artistic quality (presentation) of figure-1 should be improved.
  2. The authors should consolidate the tables to make it more readable.

Reviewer 2 Report

These are the well-thought and researched set of guidelines for the follow-up care and secondary prevention measures for the survivors of colorectal cancer. It highlights the role of the survivor's health care professionals and family/friends towards making the survivor's life better in many ways, especially when there is a relapse after the treatment. I would like the authors to address a few points: The articles and studies considered for preparing these guidelines are till 2019 only. Please consider including the information from the studies of 2020 and 2021. Subsequently, add additional points or tweak the recommendations presented in these guidelines based on those. Also, please include more knowledge and specific directions for the targeted well-being activities for the survivors that are becoming increasingly common, e.g., meditation—many cancer hospitals these days equipped with meditation centers. 
